# Transferability of Diffractive Structure in the Compression Molding of Chalcogenide Glass

**DOI:** 10.3390/mi14020273

**Published:** 2023-01-20

**Authors:** Byeong-Rea Son, Ji-Kwan Kim, Young-Soo Choi, Changsin Park

**Affiliations:** 1Department of Automotive Engineering, Honam University, Gwangju 62399, Republic of Korea; 2Department of Mechanical Convergence Engineering, Gwangju University, Gwangju 61743, Republic of Korea

**Keywords:** infrared optical lens, chalcogenide glass, diffractive structure, compression molding

## Abstract

This study investigates the use of Ge_28_Sb_12_Se_60_ chalcogenide glass for the compression molding of an infrared optical lens with a diffractive structure. Firstly, a mold core was prepared through ultra-precision grinding of tungsten carbide, and a chalcogenide glass preform was crafted through a polishing process and designed with a radius that would prevent gas isolation during the molding process. The test lens was then molded at various temperature conditions using the prepared mold core and preform. The diffractive structures of both the mold core and the resulting molded lens were analyzed using a microscope and white light interferometer. The comparison of these diffractive structures revealed that the molding temperature had an effect on the transferability of the diffractive structure during the molding of the chalcogenide glass lens. Furthermore, it was determined that, when the molding temperature was properly adjusted, the diffractive structure of the core could be fully transferred to the surface of the chalcogenide lens. Optimized chalcogenide glass-based lenses have the potential to serve as cost-effective yet high-performance IR optics.

## 1. Introduction

Recently, thermal imaging applications are being used more frequently in various high-tech industrial fields, such as automotive, medical, security, and measurement. These applications require infrared optics, which have traditionally been made through single-point diamond turning (SPDT) using crystalline materials such as germanium, zinc selenide, zinc sulfide, and silicon, which are capable of transmitting far-infrared wavelengths [1,2,3]. These crystalline materials have proven to be effective in creating high-quality infrared optics, but the SPDT process can be expensive and is not well-suited to high volume production. As a result, researchers are looking for alternative materials and methods for a mass-produced infrared lens that is both cost-effective and able to meet the demands of various industrial applications. To reduce the high costs often associated with producing infrared optics through traditional methods, researchers are currently exploring the use of chalcogenide glasses for molded infrared optics [4,5,6,7]. These glasses can be mass-produced at a lower cost than crystalline materials through the precision glass molding (PGM) process, making them a potentially attractive option for various industrial applications. Chalcogenide glasses also have unique optical properties that make them a viable alternative to crystalline materials for infrared optics [8,9,10,11]. However, more research is needed to fully understand the capabilities and potential of chalcogenide glasses for molded infrared optics as well as to optimize the production process for these materials.

Chalcogenide glasses have a high level of dispersion, with an Abbe number value that is approximately 10 times lower than that of germanium. This leads to larger chromatic aberration when chalcogenide glasses are used to make lenses. To address this issue, a diffractive surface can be incorporated into the lens design. This diffractive surface has negative dispersion properties and is able to correct chromatic aberration without the need for additional lenses. The negative dispersion property of a diffractive surface means that long-wave light is more effectively refracted than short-wave light, allowing for the compensation of chromatic aberration in chalcogenide glass lenses. The molding of diffractive-aspheric lenses using chalcogenide glass offers the possibility of creating inexpensive infrared optics with excellent optical performance. However, further research is required to understand the various parameters, such as temperature, pressure, and time, that influence the formation of the diffractive structure on the lens surface during the compression molding process. In this study, the researchers aimed to understand how molding temperature influences the transferability of the diffractive structure during the compression molding of Ge_28_Sb_12_Se_60_ chalcogenide glasses. To do this, they first machined a diffractive structure onto the surface of a mold core using the cross-grinding method. Then, using the prepared mold core, they molded a chalcogenide glass lens at various temperatures. To evaluate the transferability of the diffractive structure, the researchers used a microscope and white light interferometer to compare the diffractive structure on the mold core to the one on the resulting molded chalcogenide glass lens. Through this process, they were able to determine the effect of molding temperature on the transferability of the diffractive structure during the compression molding process.

## 2. Experiments and Discussion

### 2.1. Optical Design of the Diffractive-Aspheric Lens

The lens used in this study was a 21 mm diameter hybrid lens, as shown in Figure 1, with an aspheric and a diffractive-aspheric surface. The aspheric form is expressed as the sum of the cone and polynomial terms shown in the equation:(1)$$z=\frac{C\cdot x^{2}}{1+\sqrt{1-\left(1+K\right)\cdot C^{2}\cdot x^{2}}}+\overset{n}{\underset{i=1}{\sum }}A_{i}\cdot x^{2i}$$
where *K* is a conic constant; *C* is calculated using the equation *C = 1/R*, where *R* is the vertex radius of the aspheric surface; and *A* is a constant for the aspheric form. The diffractive structure was designed in the form of a rotationally symmetric kinoform, and the phase profile is expressed as shown in the equation:(2)$$\Phi \left(r\right)=m\frac{2\pi }{\lambda_{0}}\underset{n=1}{\sum }C_{n}r^{2n}$$
where *m* is the diffraction order; *λ*_0_ is the design wavelength; *C_n_* is the phase factor; and *r* is the radius of the diffraction ring. In this study, there are three diffraction rings with a blade depth of 6.226 µm on a concave surface, as shown in Figure 1, designed with *m* = 1 and *λ*_0_ = 10 µm, *n* = 1~4, *C*_2_ = −5.704e^−4^, *C*_4_ = −5.764e^−7^, *C*_6_ = 3.145e^−8^, and *C*_8_ = −2.168e^−10^. The diffraction radii (*r*) were 4.3 mm, 6.3 mm, and 8 mm, respectively.

### 2.2. Mold Fabrication and Molding of Diffractive-Aspheric Lens

Tungsten carbide (WC, FB01, DIJET Industrial Co., Yokohama, Japan) with cobalt (Co) content of 1.0 wt% was used to build the mold core for the molding of chalcogenide glass. The mold core’s surface was ground and then polished to improve roughness. The diffractive structure was machined on the top core of the mold by the cross-grinding method. A diamond-like-carbon (DLC) thin film of 150 nm in thickness was coated to prevent glass adhesion on the mold core’s surface and to improve the lifetime of the mold. The composition of the chalcogenide glass used in this study was Ge_28_Sb_12_Se_60_ (IRG25, SCHOT Co., Mainz, Germany); its properties are indicated in Table 1 [12,13,14].

Figure 2 shows a preform drawing designed for lens molding. If the curvature radius of the preform is greater than that of the lens, gas may become isolated on the lens surface and an air gap will occur, which is detrimental to optics. Therefore, the curvature radius of the preform was designed to be smaller than that of the lens to release the air.

The diffractive-aspheric lenses were molded by using a precision glass-molding machine (LMR-3300V2S, Daeho Technology Korea Co., Changwon, Republic of Korea). The molding process of the LMR-3300V2S consists of five major steps: preheating—heating—pressing—gradual cooling—steep cooling, as shown in Figure 3. At the preheating step, the mold’s assembly, including the chalcogenide glass preform, is heated from the bottom plate under no load, and then this assembly is heated by both top and bottom plates in the heating step. In the pressing step, the heated glass preform is pressed with constant pressure in the pressing step, and the pressed lens is slowly cooled to release the induced stress inside the lens in the gradual cooling step. Finally, the lens is rapidly cooled by using N_2_ gas and is then released from the mold. Table 2 summarizes the details of the process parameters and molding conditions used in this study. The temperature of the top plate in the pressing step was set as a variable (320~335 °C) to verify the effects on the transcription properties of the diffractive structure. Previous studies reported the effect of molding temperature in the molding of chalcogenide glass lenses [15,16,17,18,19,20]. Referring to previous work, the temperature of the preheating step was determined to be lower than the transition temperature (T_g_) of chalcogenide glass (IRG25) to prevent lens breakage. In addition, the temperatures of both the heating and the press steps were set to be higher than the softening point (T_s_) of chalcogenide glass (IRG25). Since the diffractive structure was on the top core of the mold, the temperature of the bottom plate was fixed at 320 °C during the molding process, and the temperature of the top plate was set at 5 °C intervals from 320 to 335 °C. Five lenses were molded under each molding condition. To verify the transcription properties of the diffractive structure in the molding process, the surfaces of both the mold core and the molded lens were measured using a microscope and a white light interferometer (NewView 5000, Zygo Co., Middlefield, CT, USA) and were compared.

### 2.3. Results and Discussion

Figure 4 shows the figure of the top core and a 3D image of the diffractive structure on the core surface measured with a white light interferometer. The blade depth of the diffractive structure was measured at about 6 µm, and it was confirmed that the diffractive structure of the core was machined to match the design value of 6.226 µm. Figure 5a shows the chalcogenide glass lens molded under condition *A* (320 °C for the top plate temperature) in Table 2. Breakage of the molded lens or adhesion of the glass to the mold core surface was not found, and only fine surface defects were observed. Chien et al., in a previous study, reported the cause of these defects [21]. The defects are thought to be derived from the evaporation of volatile elements, such as Sb, and were presumed to pose no serious problem to LWIR (8~12 µm) transmittance of a chalcogenide glass lens.

Figure 5b shows a microscopic image and a 3D image of the diffractive structure on the molded lens. It was found that the diffractive structure of the mold core surface was not normally transferred and that its tendency was worse as it grew from the center to the outside. This fault is assumed to be caused by an air gap in the inner edge of the diffractive structure during the compression of the chalcogenide glass, as shown in Figure 6. When the glass material cannot be pushed into the inner edge of the diffractive structure during the pressing step, an air gap is generated in the corresponding part and causes deterioration in lens performance.

Figure 7 shows a comparison of the diffractive structure on the molded lens surface molded under each condition, from *A* to *D*, in Table 2. When the temperature of the top plate in the pressing step was raised to 325 °C (condition *B*), the first diffraction ring on the inside area was transferred completely, and the transcription of the second diffraction ring was performed at 330 °C (condition *C*). All diffraction rings of the core surface were fully transferred to the lens surface at 335 °C (condition *D*). Both the applied force and compression time were reduced from the center of the molded lens to the outside. For this reason, it is assumed that the transferability of the inner first diffraction ring is the best and that further towards the outside, it becomes worse. In addition, the experimental results show that the diffractive structure can be transferred without an air gap when the chalcogenide glass is sufficiently softened during the compression. Therefore, when the molding temperature is increased to lower the viscosity of the material for forming the diffractive lens, the transfer characteristics are improved with the diffractive structure.

However, at a high molding temperature, serious surface defects on the lens surface or material fusion on the mold core surface may occur due to volatile substances generated from the chalcogenide glass material. The form error of the lens molded under condition *D* was shown in Figure 8. All diffractive structure of the mold core was confirmed to be perfectly transferred to the surface of the chalcogenide glass lens.

## 3. Conclusions

The objective of this study was to investigate the potential of using Ge28Sb12Se60 chalcogenide glass to create a diffractive-aspheric lens through compression molding. To this end, we first prepared a mold core with a diffractive structure and used it to mold the lens at various temperature conditions. The experimental results showed that the diffractive structure could be transferred onto the chalcogenide glass without the presence of an air gap as long as the glass was sufficiently softened during the compression process. Additionally, we found that the transferability of the inner diffraction ring was the most successful, with decreasing success as the rings moved further outward. This phenomenon is believed to be due to the combination of the force applied and the duration of the compression. Optimized chalcogenide glass-based lenses have the potential to be used as inexpensive yet high-performance IR optics.

## Figures and Tables

**Figure 1 micromachines-14-00273-f001:**
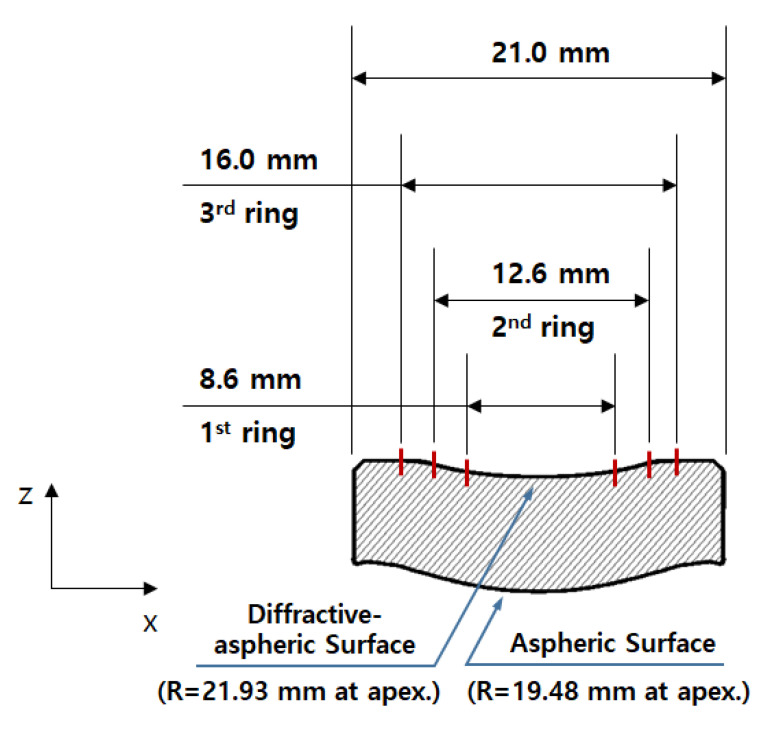
Drawing of the diffractive-aspheric lens.

**Figure 2 micromachines-14-00273-f002:**
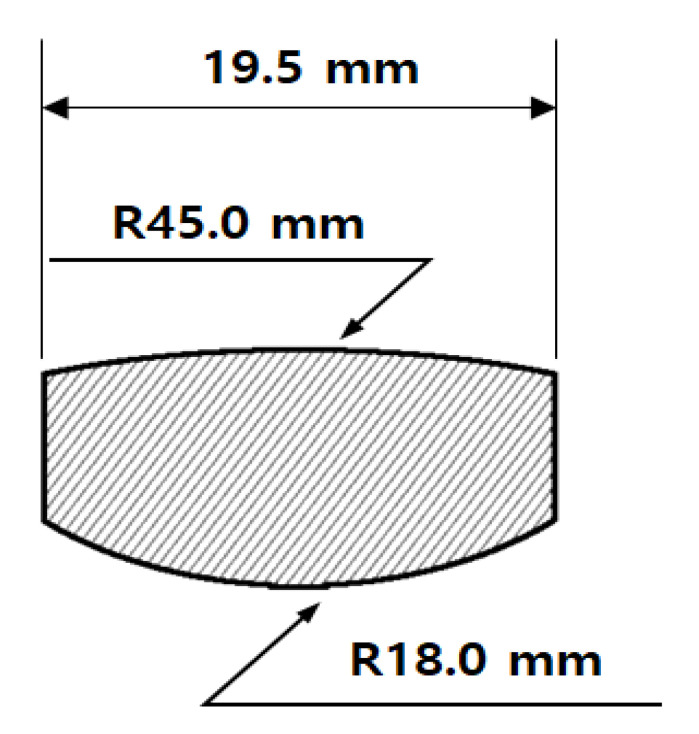
Drawing of the preform for molding of the diffractive-aspheric lens.

**Figure 3 micromachines-14-00273-f003:**
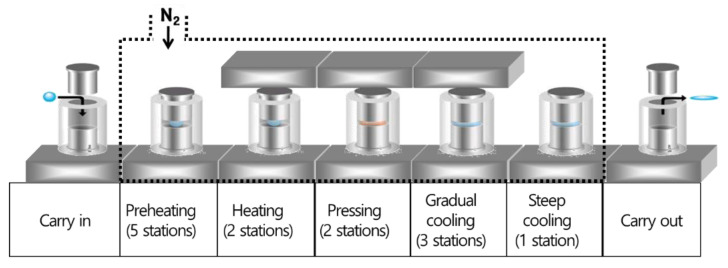
Schematic diagram of the molding process in LMR-3300V2S.

**Figure 4 micromachines-14-00273-f004:**
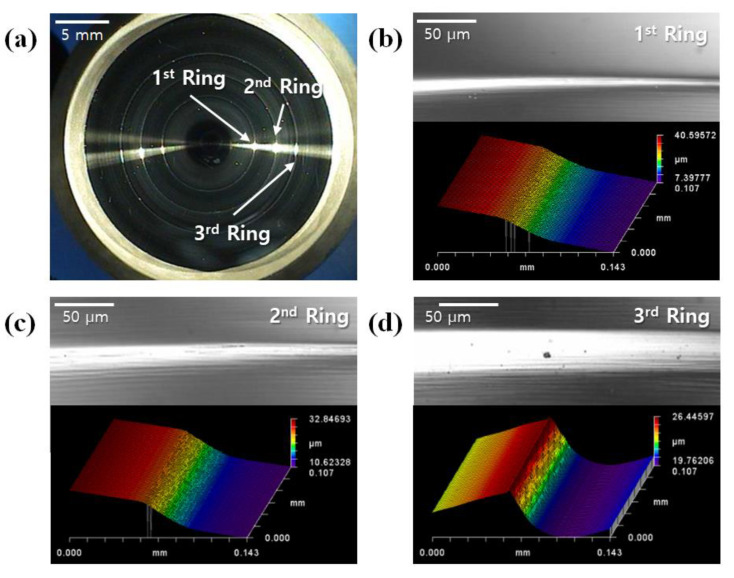
Diffractive-aspheric surface of the mold core: (**a**) microscopic image of the entire surface and (**b**–**d**) microscopic image and 3D surface view of diffraction rings.

**Figure 5 micromachines-14-00273-f005:**
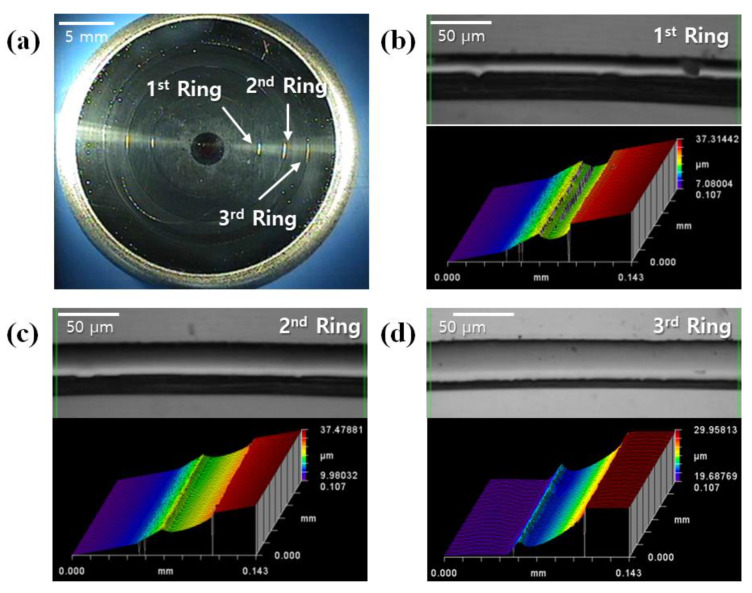
Diffractive-aspheric surface of the chalcogenide glass lens molded under condition *A* (*molding temperature* 320 °C): (**a**) microscopic image of the entire surface and (**b**–**d**) microscopic image and 3D surface view of diffraction rings.

**Figure 6 micromachines-14-00273-f006:**
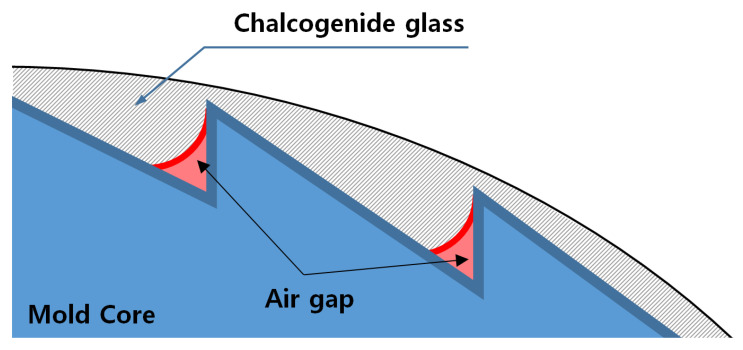
Schematic diagram of the interface between the glass and the mold core during the pressing step.

**Figure 7 micromachines-14-00273-f007:**
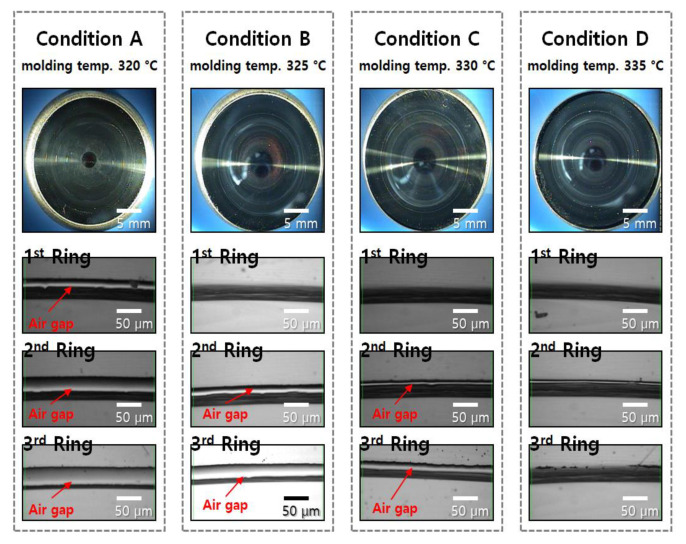
Diffractive-aspheric surface image and microscopic image of the diffraction ring in the molded chalcogenide glass under the conditions *A* to *D*. Microscopic images of the diffraction ring in the molded chalcogenide glass under the conditions *A* to *D*.

**Figure 8 micromachines-14-00273-f008:**
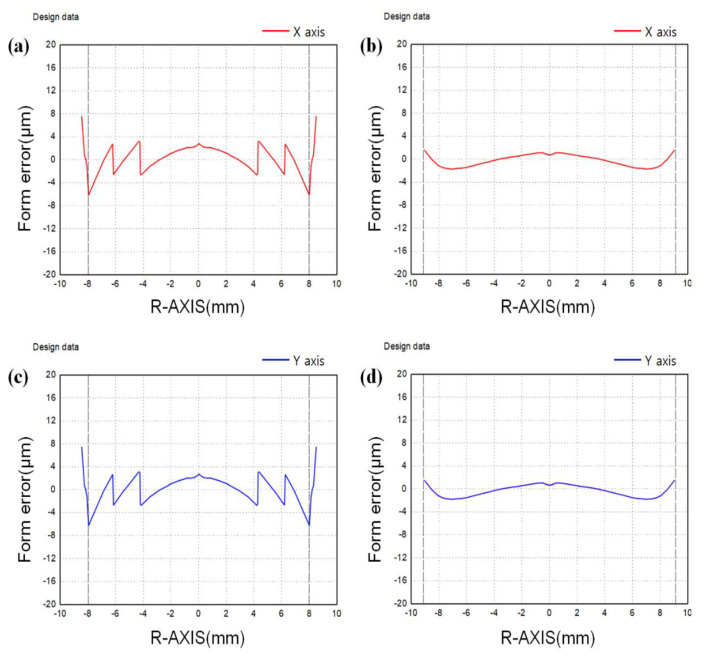
Measurement results of the lens form molded under condition *D*: (**a**) diffractive-aspheric surface (*x*-axis), (**b**) aspheric surface (*x*-axis), (**c**) diffractive-aspheric surface (*y*-axis) and (**d**) aspheric surface (*y*-axis).

**Table 1 micromachines-14-00273-t001:** Thermal properties of chalcogenide glass (IRG25).

Term	Unit	Value
Transition temperature (T_g_)	°C	285
Softening temperature (T_s_)	°C	310
Thermal conductivity	W/m·K	0.25
Thermal expansion coefficient	10^−6^/K	14

**Table 2 micromachines-14-00273-t002:** Process parameters and molding conditions used in this study.

*Process Parameters*					
		Preheating	Heating	Pressing	Gradual Cooling
Temp. (°C)	Top plate	-	320	320~335	200
	Bottom plate	280	320	320	200
Pressure (MPa)		-	-	0.2	0.05
Unit-process time (s)		300			
** *Molding Conditions* **					
Condition no.		*A*	*B*	*C*	*D*
Temp. (°C)		320	325	330	335

## Data Availability

Not applicable.

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
