# Peer review of "Transferability of Diffractive Structure in the Compression Molding of Chalcogenide Glass"

_micromachines, 2023, doi:10.3390/mi14020273_

Round 1
Reviewer 1 Report
Comments to the Authors:
In summary:
Authors investigated the use of Ge28Sb12Se60 chalcogenide glass for the compression molding of an infrared optical lens with a diffractive structure. They molded test lenses at various temperature conditions using the prepared mold core and preform. Later a microscope and a white light interferometer were used to analyze the diffractive structures of both the mold core and the resulting molded lens. Their optimized chalcogenide glass-based lenses have the potential to be used as inexpensive yet high-performing IR optics.
Before I recommend this manuscript for publication in Micromachines, several issues should be clarified, listed as below:
1. Could the authors provide the detailed value of the Cn and order n in equation (2)?
2. In Ref [21], the transmittance of 2~6um wavelength was measured, but not for LWIR (8~12um). How can authors presume there is no serious problem on LWIR transmittance in Line 136 based on [21]?
3. To prove their method is cost-effective in Line 21 and inexpensive in Line185, they should compare their cost with others quantitatively, like providing a table or graph.
4. Minor corrections:
a. It seems marked wrong in Figure 2. The top should be R45.0mm and the bottom should be R18.0mm.
b. The temperatures of Molding Conditions in Table 2 is described at Page 7. To make the paper more readable, it's better to separate this part from Table 2 as a new table to be put at Page 7.
c. In Line 111, "A previous study" should be corrected as " Previous studies" since there are 6 references ([15-20]).
Author Response
Manuscript Title: Transferability of Diffractive Structure in the Compression Molding of Chalcogenide Glass
Manuscript Number: micromachines-2160414
Response comment We sincerely appreciate the Editor’s strong support, the Reviewer’s appreciation, and very positive feedback on the quality of our manuscript. As recommended by the Editor, we have thoroughly reviewed the decision letter and the comments provided by the Reviewers. The Reviewer's words of encouragement and unequivocal support inspire us to pursue our research endeavors with a lot of confidence. Herewith, we are submitting a revised version of our manuscript with explanations to the observations to accommodate all the comments raised by the Reviewers. The details of the changes made in the revised version of the manuscript are provided as a marked copy as desired. We sincerely hope the details provided along with the suggested modifications justify our conclusions and would help us to get an opportunity to publish our work in your esteemed journal “Micromachines”. We hope that the Editorial desk would now find everything in order.
Point-by-Point Response to the Reviewer’s Comments
Reviewer 1 Comments
Authors investigated the use of Ge28Sb12Se60 chalcogenide glass for the compression molding of an infrared optical lens with a diffractive structure. They molded test lenses at various temperature conditions using the prepared mold core and preform. Later a microscope and a white light interferometer were used to analyze the diffractive structures of both the mold core and the resulting molded lens. Their optimized chalcogenide glass-based lenses have the potential to be used as inexpensive yet high-performing IR optics.
Before I recommend this manuscript for publication in Micromachines, several issues should be clarified, listed as below
Comment 1. Could the authors provide the detailed value of the Cn and order n in equation (2)?
Response: We sincerely thank the reviewer for his valuable suggestions. As recommended by the reviewer, we have done our best to rectify providing in detail the equation (2) as below.
(Previous version of the manuscript) In this study, there are three diffraction rings with a blade depth of 6.226 µm on a concave surface, as shown in Fig. 1, designed with m = 1 and λ0 = 10 µm. The diffraction radius (r) is 4.3 mm, 6.3 mm, and 8 mm, respectively.
(Revised version of the manuscript) In this study, there are three diffraction rings with a blade depth of 6.226 µm on a concave surface, as shown in Fig. 1, designed with m = 1, λ0 = 10 µm, n=1~4, C2=-5.704e-4, C4=-5.764e-7, C6=3.145e-8, and C8=-2.168e-10. The diffraction radius (r) is 4.3 mm, 6.3 mm, and 8 mm, respectively.
Comment 2. In Ref [21], the transmittance of 2~6um wavelength was measured, but not for LWIR (8~12um). How can authors presume there is no serious problem on LWIR transmittance in Line 136 based on [21]?
Response: We sincerely thank the reviewer for his valuable Question. Since the wavelength of the LWIR (8-12 um) is longer than the wavelength (2-6 um) used in Ref[21], it is judged that the transmittance will not be a problem. In general, the shorter the wavelength of light used when measuring transmittance, the lower the transmittance value due to surface defects.
Comment 3. To prove their method is cost-effective in Line 21 and inexpensive in Line185, they should compare their cost with others quantitatively, like providing a table or graph.
Response: We sincerely thank the reviewer for his valuable suggestions. As recommended by the reviewer, we have done our best to provide the clearest sentence in the revised version of the manuscript.
(Previous version of the manuscript) As a result, researchers are looking for alternative materials and methods for creating infrared optics that are both cost-effective and able to meet the demands of various industrial applications.
(Revised version of the manuscript) As a result, researchers are looking for alternative materials and methods for a mass-producing infrared lens that is both cost-effective and able to meet the demands of various industrial applications.
Comment 4. Minor corrections:
a. It seems marked wrong in Figure 2. The top should be R45.0mm and the bottom should be R18.0mm.
Response: We sincerely regret for our oversight. The suggested line has been rectified in the revised version of the manuscript.
b. The temperatures of Molding Conditions in Table 2 is described at Page 7. To make the paper more readable, it's better to separate this part from Table 2 as a new table to be put at Page 7.
Response: We sincerely thank the reviewer for his valuable suggestion. The suggested line has been modified in the revised version of the manuscript.
c. In Line 111, "A previous study" should be corrected as " Previous studies" since there are 6 references ([15-20]).
Response: Complied with the reviewer’s suggestion.
(Previous version of the manuscript) A previous study reported the effect of molding temperature in the molding of chalcogenide glass lenses [15-20].
(Revised version of the manuscript) Previous studies reported the effect of molding temperature in the molding of chalcogenide glass lenses [15-20].

Reviewer 2 Report
Reviewer Comments
Manuscript Title: Transferability of Diffractive Structure in the Compression Molding of Chalcogenide Glass
Manuscript Number: micromachines-2160414
The manuscript under review is devoted to studying the use of chalcogenide glass for the compression molding of an infrared optical lens with a diffractive structure. Researchers are looking for alternative materials and methods for creating infrared optics that are cost-effective. In this paper, the researchers aimed to understand how molding temperature influences the transferability of the diffractive structure during the compression molding of chalcogenide glasses.
The manuscript contains new and significant. The abstract clearly and accurately describes the content of the article. The literature review part contains distinct and rich references. The paper is nicely written and can be accepted but first, it should be improved. I have these comments:
1- In line 56, I think that the word “impacts” should be replaced by the word “influences”.
2- In the title of figure 5, it will be better if you indicate the temperature of the condition A.
3- Can you more explain the figure 6 and the role of the air gaps ?
4- In figure 8, the curve of the Y-Axis is not available. Can you explain?
Finally, I recommend the paper for publication after resolving the minor comments.

Author Response
Manuscript Title: Transferability of Diffractive Structure in the Compression Molding of Chalcogenide Glass
Manuscript Number: micromachines-2160414
Response comment We sincerely appreciate the Editor’s strong support, the Reviewer’s appreciation, and very positive feedback on the quality of our manuscript. As recommended by the Editor, we have thoroughly reviewed the decision letter and the comments provided by the Reviewers. The Reviewer's words of encouragement and unequivocal support inspire us to pursue our research endeavors with a lot of confidence. Herewith, we are submitting a revised version of our manuscript with explanations to the observations to accommodate all the comments raised by the Reviewers. The details of the changes made in the revised version of the manuscript are provided as a marked copy as desired. We sincerely hope the details provided along with the suggested modifications justify our conclusions and would help us to get an opportunity to publish our work in your esteemed journal “Micromachines”. We hope that the Editorial desk would now find everything in order.
Point-by-Point Response to the Reviewer’s Comments
Reviewer 2 Comments
The manuscript under review is devoted to studying the use of chalcogenide glass for the compression molding of an infrared optical lens with a diffractive structure. Researchers are looking for alternative materials and methods for creating infrared optics that are cost-effective. In this paper, the researchers aimed to understand how molding temperature influences the transferability of the diffractive structure during the compression molding of chalcogenide glasses.
The manuscript contains new and significant. The abstract clearly and accurately describes the content of the article. The literature review part contains distinct and rich references. The paper is nicely written and can be accepted but first, it should be improved. I have these comments:
Comment 1. In line 56, I think that the word “impacts” should be replaced by the word “influences”.
Response: Complied with the reviewer’s suggestion
(Previous version of the manuscript) In this study, the researchers aimed to understand how molding temperature impacts the transferability of the diffractive structure during the compression molding of Ge28Sb12Se60 chalcogenide glasses.
(Revised version of the manuscript) In this study, the researchers aimed to understand how molding temperature influences the transferability of the diffractive structure during the compression molding of Ge28Sb12Se60 chalcogenide glasses.
Comment 2. In the title of figure 5, it will be better if you indicate the temperature of the condition A.
Response: Complied with the reviewer’s suggestion
(Previous version of the manuscript) Figure 5. Diffractive-aspheric surface of the chalcogenide glass lens molded under the condition A: (a) microscopic image of the entire surface and (b-d) microscopic image and 3D surface view of diffraction rings.
(Revised version of the manuscript) Figure 5. Diffractive-aspheric surface of the chalcogenide glass lens molded under the condition A (molding temperature 320 °C): (a) microscopic image of the entire surface and (b-d) microscopic image and 3D surface view of diffraction rings.
Comment 3. Can you more explain the figure 6 and the role of the air gaps ?
Response: We sincerely thank the reviewer for his valuable suggestions.
When the glass material cannot be pushed into the inner edge of the diffractive structure during the pressing step, an air gap is generated in the corresponding part and they cause deterioration of lens performance.
Comment 4. In figure 8, the curve of the Y-Axis is not available. Can you explain?
Response: We sincerely thank the reviewer for his valuable suggestions. As recommended by the reviewer, we have divide the figure 8. With X-axis and Y-axis. As reviewer knowns, The lens has same the geomatic and dimensions of the X and Y axis. Previous figure was shown the same axis within the X and Y axis in the graph. From the reviewer suggestion, we changed figure 8 and captions as below.
(Previous version of the manuscript) Fig. 8. Measurement result of the lens form molded under condition D: (a) diffractive-aspheric surface and (b) aspheric surface
(Revised version of the manuscript) Fig. 8. Measurement result of the lens form molded under condition D: (a) diffractive-aspheric surface(x-axis), (b) aspheric surface(x-axis), (c)diffractive-aspheric surface(y-axis) and (d) aspheric surface(y-axis)
